# The Efficacy of PD-1/PD-L1 Inhibitors in Patients with Liver Metastasis of Non-Small Cell Lung Cancer: A Real-World Study

**DOI:** 10.3390/cancers14174333

**Published:** 2022-09-05

**Authors:** Mingying Xie, Na Li, Xiaoling Xu, Yanjun Xu, Hui Li, Liang Zhu, Jiamin Sheng, Zichao Zhou, Yun Fan

**Affiliations:** 1Department of Oncology, The Second Clinical Medical College, Zhejiang Chinese Medical University, Hangzhou 310053, China; 2Department of Medical Oncology, Cancer Hospital of the University of Chinese Academy of Sciences (Zhejiang Cancer Hospital), Hangzhou 310022, China; 3Department of Oncology, The First Clinical Medical College, Wenzhou Medical University, Wenzhou 325000, China; 4Institute of Cancer and Basic Medicine (ICBM), Chinese Academy of Sciences, Hangzhou 310022, China; 5Department of Pathology, Zhejiang Cancer Hospital, Hangzhou 310022, China

**Keywords:** liver metastasis, non-small cell lung cancer, PD-L1, PD-1/PD-L1 inhibitor, CD8+ T cell

## Abstract

**Simple Summary:**

The liver is a common metastatic site of non-small cell lung cancer (NSCLC) and is associated with a poor prognosis. Immune checkpoint inhibitors (ICIs), represented by programmed death-1 (PD-1)/ programmed death ligand-1 (PD-L1) inhibitors, have significantly improved efficacy in patients with advanced NSCLC, but the efficacy in patients with NSCLC and liver metastases remains controversial. We aimed to evaluate the efficacy of PD-1/PD-L1 inhibitors in patients with NSCLC and liver metastases in the real-world. In this study, we illustrated that PD-1/PD-L1 inhibitors are effective in NSCLC patients with liver metastases but were inferiorly effective in patients without liver metastases. In addition, PD-L1 expression and CD8+ T cell infiltration may be potential biomarkers for PD-1/PD-L1 inhibitor therapy in NSCLC patients with liver metastases.

**Abstract:**

Background: A controversy exists regarding the efficacy of programmed death-1 (PD-1)/ programmed death ligand-1 (PD-L1) inhibitors for patients with non-small cell lung cancer (NSCLC) and liver metastases. Our study retrospectively evaluated the efficacy of PD-1/PD-L1 inhibitors in NSCLC patients with liver metastases. Methods: This retrospective study included 1627 lung cancer patients who received immunotherapy. Among 648 patients who had advanced NSCLC and received PD-1/PD-L1 inhibitors, 61 had liver metastases and 587 did not have. We analyzed patient characteristics, progression-free survival (PFS) and overall survival (OS). An exploratory analysis of biomarkers including CD4, CD8 and CD68 for efficacy in patients with liver metastases was also performed. Results: In liver metastasis patients receiving PD-1/PD-L1 inhibitors, the objective response rate (ORR) was 29.5%, the disease control rate (DCR) was 72.1%, PFS was 6.4 months and OS was 15.2 months, which were all worse than those of patients without liver metastases (ORR: 35.8%; DCR: 81.8%; PFS: 7.9 months, *p* = 0.001; OS: 20.6 months, *p* = 0.008). When compared to non-liver lesions, the ORR (26.2 vs. 39.3%) and DCR (75.4 vs. 88.5%) of liver lesions were lower. During the analysis of PD-L1 expression, 27 PD-L1-positive patients had a longer PFS than 21 patients in the negative group (*p* = 0.012). Being PD-L1 positive was the independent prognostic indicators for PFS (*p* = 0.006). Additionally, the PD-L1 and CD8 dual-positive group responded favorably to PD-1/PD-L1 inhibitors. Conclusions: PD-1/PD-L1 inhibitors are effective in liver metastasis–NSCLC patients. However, the efficacy is inferior when compared to those of patients without liver metastases. In NSCLC patients with liver metastases, PD-L1 expression and CD8+ T cell infiltration can predict the response of PD-1/PD-L1-directed immunotherapy.

## 1. Introduction

Approximately 15–20% of patients with metastatic non-small cell lung cancer (NSCLC) present with liver metastasis, and liver metastasis correlates with an inferior prognosis [1,2]. Throughout the chemotherapy era, the median overall survival (OS) in patients with NSCLC and liver metastasis was fewer than ten months [3,4,5]. Metastasis to the liver is an independent negative predictor for NSCLC patients who receive chemotherapy [6,7].

For advanced NSCLC patients, immune checkpoint inhibitors (ICI) such as programmed cell death-1 (PD-1)/programmed cell death ligand-1 (PD-L1) inhibitors have been established as first-line therapy regimen [8,9]. Owing to the complicated microenvironment of metastatic liver lesions, the efficacy of ICIs in NSCLC patients with liver metastases has recently attracted much attention. Recently, preclinical mouse models discovered that liver metastasis drained activated CD8+ T cells from the systemic circulation and triggered CD8+ T cell death, resulting in a systemic immunological desert, implying that immunotherapy effectiveness for NSCLC patients with liver metastasis is diminished [10]. In clinical situations, it still remains controversial whether ICIs are effective in NSCLC patients with liver metastasis. A subgroup analysis in several phase III clinical studies has shown that NSCLC patients with liver metastases can benefit from ICI therapy [3,4]. In addition, other studies have reported that the efficacy of ICIs in NSCLC patients with liver metastases is similar to that in patients without liver metastases [11,12]. Conversely, several studies have drawn different conclusions, suggesting that ICI has limited efficiency in NSCLC patients who have liver metastasis. Data in KEYNOTE-001 showed that progression-free survival (PFS) from NSCLC patients with liver metastases after treatment with pembrolizumab was significantly shorter (*p* < 0.05), and the objective response rate (ORR) was lower (28.6 vs. 56.7%) than patients without liver metastases [13]. Several retrospective studies have drawn similar conclusions [14,15,16].

On the basis of the available evidence from subgroup analyses of clinical studies and retrospective studies, the efficacy of anti-PD-1/PD-L1 therapy in patients with NSCLC and liver metastases remains controversial. Therefore, we conducted this retrospective study in order to evaluate the effectiveness for PD-1/PD-L1 inhibitors on liver metastasis-NSCLC patients. We further enrolled NSCLC patients without liver metastases who received ICI therapy during the same period to compare their efficiency. We also explored potential prognostic factors. In addition, evaluation was carried out on the expression of PD-L1 and the infiltration of immune cells (IC).

## 2. Materials and Methods

### 2.1. Patients

A total of 1627 patients who had lung cancer and who received PD-1/PD-L1 inhibitors at Zhejiang Cancer Hospital from September 2017 to December 2020 were screened. Of these patients, 61 patients with NSCLC and liver metastasis were enrolled. Simultaneously, 587 NSCLC patients without liver metastasis were selected as comparators. The major inclusion criteria were: Eastern Corporation Oncology Group performance status (ECOG PS) of 0–2, cytologically or histologically confirmed stage IV NSCLC, at least one measurable lesion, treatment with PD-1/PD-L1 inhibitors, and complete survival data. The major exclusion criteria were: small-cell lung cancer, sensitive epidermal growth factor receptor (EGFR) mutations, anaplastic lymphoma kinase (ALK) rearrangements or ROS proto-oncogene 1 receptor tyrosine kinase (ROS1) fusions, and incomplete follow-up data. According to the Response Evaluation Criteria in Solid Tumors (RECIST) version 1.1, the response was evaluated radiographically [17]. The PFS was calculated as the time between starting PD-1/PD-L1 inhibitors and the progression of disease or the incidence of death owing to different causes. The OS was defined as the period from the initiation of PD-1/PD-L1 inhibitor treatment until death due to any cause or the latest follow-up date. These enrolled patients were followed up until June 2021. This study was approved by the Zhejiang Cancer Hospital’s Ethics Committee.

### 2.2. Immunohistochemistry (IHC)

We obtained tumor tissues from different origins, including the lung, liver, lymph nodes, soft tissue and bones. According to manufacturer’s recommendations, IHC was conducted using 4–5 μm formalin-fixed and paraffin-embedded (FFPE) sections. The PD-L1 clone 28-8 pharmDx kit and the Dako Automated Link 48 platform were used to measure the expression of PD-L1. CD4 (clone B468A, diluted at 1:200, Santa Cruz, Texas, USA) and CD8 (clone 144B, diluted at 1:100, Abcam, Cambridge, UK) expression in T cells, and CD68 (clone PG-M1, diluted at 1:600, Abcam, Cambridge, UK) expression on macrophages were also assessed. Two pathologists independently scored the stained tissues, and the clinical parameters were kept confidential from these two pathologists.

The tumor proportion score (TPS) was used to assess PD-L1 expression. The percentage of tumor cells stained with partial or complete membranes was the definition of TPS. The positive expression of PD-L1 was considered as TPS ≥ 1%. Among all nucleated cells in the tumor mesenchyme, the proportion of positive cells for CD4, CD8, and CD68 expression was assessed. Positivity on lymphocytes was set at 5, 5, and 20% for CD4, CD8, and CD68, respectively. Based on PD-L1 expression and CD8+ T cell infiltration, the tumor microenvironment was divided into three subgroups (both negative, both positive, or single-positive).

### 2.3. Statistical Analyses

SPSS (version 25.0) and GraphPad Prism (version 9.2.0) were used to perform the statistical analysis. The distribution of clinical features was assessed through the Chi-squared test. The responses were graded based on Response Evaluation Criteria in Solid Tumors (RECIST) version 1.1 [17]. Kaplan–Meier curves were applied for survival analysis of PFS and OS in various groups. Univariate and multivariate Cox regressions were used for prognostic analysis to obtain the adjusted hazard ratios (HR) and 95% confidence intervals (95% CI). The multivariate Cox regression analysis obtained variables with a *p* < 0.10 in the univariate analysis. Statistical significance was set of *p* < 0.05, and two-tailed *p* values were calculated in all reports.

## 3. Results

### 3.1. Patient Characteristics

Sixty-one NSCLC patients who had liver metastasis were enrolled in this study (Figure A1). They were all treated with anti-PD-1/PD-L1 therapy. The median age of these patients was 63 years (range, 34–76 years). Among them, 59.0% (*n* = 36) were < 65 years old, 90.2% (*n* = 55) were male, 86.9% (*n* = 53) had a ECOG PS of 0–1, 82.0% (*n* = 50) were current or former smokers, and 11.5% (*n* = 7) had brain metastasis. The histology of the tumors was 27 (44.3%) adenocarcinomas, 29 (47.5%) squamous cell carcinomas, and 5 (8.2%) other NSCLCs. In 55.7% (*n* = 34) of the patients, PD-1/PD-L1 inhibitors were used as a first-line therapy, and 34.4% (*n* = 21) of the patients received them as monotherapy. Meanwhile, the control group consisted of 587 metastatic NSCLC patients who were free of liver metastasis treated with PD-1/PD-L1inhibitors. The majority of patients (60/61,98.4%) were treated with PD-1 inhibitors, of whom 14 (23.0%) were treated with Pembrolizumab, 8 (13.1%) with Nivolumab, 11 (18.0%) with Tislelizumab, 5 (8.2%) with Toripalimab, 11 (18.0%) with Sintilimab, and 11 (18.0%) with Camrelizumab; only one patient (1.6%) was treated with a PD-L1 inhibitor (Atezolizumab). Table 1 shows the baseline characteristics of these patients according to the liver metastasis status. Patients who had liver metastasis experienced a considerably lower positive rate of PD-L1 expression (*p* = 0.012). There was no statistically significant difference between the liver metastasis and non-liver metastasis groups in terms of age, gender, ECOG PS, smoking, brain metastasis, histological type, lines of ICI and ICI treatment regimens. The ICI treatment regimens included ICI monotherapy and ICI combined therapy (combined chemotherapy or combined anti-angiogenesis therapy).

### 3.2. Efficacy of PD-1/PD-L1 Inhibitors

Among 61 patients who had NSCLC and liver metastasis, none achieved a complete response (CR), 29.5% (*n* = 18) showed a partial response (PR), 42.6% (*n* = 26) showed stable disease (SD), and 27.9% (*n* = 17) had progressive disease (PD) (Table 2a). In these patients, the ORR was 29.5%, the disease control rate (DCR) was 72.1% (Table 2a), PFS was 6.4 months (Figure 1a) and OS was 15.2 months (Figure 1b). Of the non-liver metastasis patients, 0.3% (*n* = 2) achieved CR, 35.4% (*n* = 208) achieved PR, 46.0% (*n* = 270) showed SD, and 18.2% (*n* = 107) had PD. The ORR and DCR of these patients were 35.8 and 81.8%, respectively (Table 2a), the PFS was 7.9 months (Figure 1c) and OS was 20.6 months (Figure 1d). Liver metastasis patients had lower ORR and DCR, and a significantly shorter PFS (*p* = 0.001) and OS (*p* = 0.008).

To further explore whether the effectiveness in PD-1/PD-L1 inhibitors on liver metastatic lesions was distinctive from other lesions, we evaluated the response rate of liver and non-liver lesions, and we found that the ORR (26.2 vs. 39.3%) and DCR (75.4 vs. 88.5%) were lower for liver lesions (Table 2b).

### 3.3. Exploratory Analysis of the Expression of Tumor Immune Microenvironment-Related Markers

In our study, 48 patients with NSCLC with liver metastasis were evaluated for PD-L1 expression (Figure A2a,b were representative images of positive and negative PD-L1 staining) and 42 patients with NSCLC with liver metastasis were tested for CD4, CD8 and CD68 infiltrates. We discovered that CD8+ T cell infiltration was considerably lower in liver lesions than that in non-liver lesions (70.0 vs. 93.8%, *p* = 0.043). Among the PD-L1 expression (60.0 vs. 55.3%, *p* = 0.788), CD4+ T cell infiltration (70.0 vs. 56.2%, *p* = 0.439), and CD68+ T cell infiltration (90.0 vs. 78.1%, *p* = 0.404), there was no statistically significant difference between liver lesions and non-liver lesions (Table 3).

### 3.4. Correlation Analysis between Biomarkers of Tumor Immune Microenvironment and Efficacy

In liver metastasis patients, the PFS in the PD-L1-positive expression group was statistically substantially prolonged when compared to the negative group (8.2 vs. 4.4 months, *p* = 0.012) (Figure 2a), while the OS was similar (*p* = 0.587) (Figure 2b). For the tumor immune microenvironment subgroups, the PD-L1 and CD8 double-positive group had the longest PFS (8.7 months, *p* = 0.004) (Figure 2e) but no difference in OS among three groups (*p* = 0.684) (Figure 2f).

Between the CD8-positive and -negative groups, no statistical difference was observed in PFS (6.6 vs. 2.7 months, *p* = 0.594) and OS (15.2 months vs. undefined, *p* = 0.427) (Figure 2c,d). In addition, the PFS and OS in the CD4 positive group were also similar to the negative cohort (PFS: *p* = 0.852; OS: *p* = 0.739), as well as for CD68 (PFS: *p* = 0.552; OS: *p* = 0.865).

### 3.5. Prognostic Analysis of Patients with Liver Metastases

Univariate analysis indicated that brain metastases were in related to a poor PFS, while PD-L1-positive expression was in association with a superior PFS (Table 4a). Multivariate analysis found that brain metastases (HR: 2.86, 95%CI: 1.04–7.84, *p* = 0.041) and PD-L1 expression (HR: 0.39, 95%CI: 0.20–0.76, *p* = 0.006) were independent prognostic factors (Table 4a). No factors were associated with OS in the univariate analysis (Table 4b).

## 4. Discussion

The liver is an immune-tolerant organ and the liver microenvironment is highly immunosuppressive [10,18]. Hepatocytes, Kupffer cells, liver sinusoidal endothelial cells, hepatic stellate cells and liver dendritic cells can recruit T cells through antigen presentation, which also leads to PD-L1 expression and binding to PD-1 on T cells, which preferentially leads to immune tolerance [19,20]. Therefore, immunotherapy may not be effective for liver metastatic lesions. Several translational studies have revealed that liver metastasis mediates immunotherapy resistance in liver lesions and hampers systemic antitumor immunity. Meng Qiao et al. carried out IHC on the tumor tissues of liver metastasis–NSCLC patients, and IHC revealed that liver metastasis–NSCLC patients had a lower percentage of PD-L1+CD8+ ICs in comparison to non-liver metastasis patients (0 vs. 30.8%, *p* = 0.088) [13]. Furthermore, it was reported that liver metastasis siphoned activated CD8+ T cells, which were derived from the systemic circulation. Subsequently, monocyte-derived macrophages induced the apoptosis of CD8+ T cells, creating a systemic immune desert in preclinical mouse models [10]. In addition, Lee et al. found that liver metastases induce the accumulation of Tregs and suppressive monocytes, leading to impaired systemic antitumor immunity [21]. Accordingly, this study intended to explore the efficacy of PD-1/PD-L1 inhibitors in liver metastasis–NSCLC patients and preliminarily investigate the specific immune microenvironment of these patients.

Marina C et al. [22] found that first-line chemotherapy had the ORR of 14.3% only in non-squamous NSCLC patients with liver metastasis. In comparison to the effectiveness of chemotherapy, we discovered that anti-PD-1/PD-L1 therapy improves outcomes among liver metastasis–NSCLC patients. The results of several recent large phase III clinical studies have shown that PD-1/PD-L1 inhibitors are effective in liver-metastatic NSCLC patients [3,4,5]. Our study demonstrated the advantages of PD-1/PD-L1 inhibitors for liver metastasis–NSCLC patients in the real-world situation. In addition, liver metastasis patients who were treated with PD-1/PD-L1 inhibitors had worse treatment outcomes (PFS, OS, ORR and DCR) than non-liver metastasis patients. This was subsequently confirmed in several retrospective studies [23,24]. Tumeh et al. [13] showed that when receiving pembrolizumab, liver metastasis–NSCLC patients had a significantly shorter PFS (1.8 vs. 4.0 months, *p* = 0.0094) and lower ORR (28.6 vs. 56.7%) than non-liver metastasis patients. Kitadai et al. [15] carried out a real-world retrospective study. A total of 215 advanced NSCLC patients treated with ICI monotherapy were admitted, including 41 (19.1%) with liver metastasis. This study reached similar conclusions that the patients with liver metastasis had worse OS (*p* < 0.001) and PFS (*p* < 0.001), and liver metastasis was an independent poor prognostic factor on the multifactorial analysis. One possible explanation for the limited efficacy in NSCLC patients with liver metastasis treated with PD-1/PD-L1 inhibitors is the immune microenvironment, which is specific to liver metastasis as mentioned above, and the suppression of systemic immunity by liver metastasis. Previous studies have confirmed that liver metastasis is a poor prognostic factor in NSCLC patients, and was associated with shorter PFS and OS [7,25,26]. Therefore, the inferior survival of NSCLC patients with liver metastasis treated with PD-1/PD-L1 inhibitors may be related to the characteristic of poor prognosis in liver metastasis.

Other studies did not find significant difference in ICI efficacy between NSCLC patients with liver metastasis and without. A real-world study which included 1470 patients but only contained 11 patients with liver metastasis found that PFS (*p* = 0.955) and OS (*p* = 0.968) were similar in liver metastasis patients or non-liver metastasis patients in treating with ICIs [11]. Immunochemotherapy was equally effective in lung cancer patients with or without liver metastasis (the HR of PFS: 1.11, 95%CI: 0.92–1.34; the HR of OS: 1.03, 95%CI: 0.80–1.35) from a meta-analysis [12]. However, the meta-analysis included NSCLC and small cell lung cancer and lacked direct PFS or OS data. To our knowledge, this is the first largest scale retrospective study to evaluate the efficacy of PD-1/PD-L1 inhibitors in NSCLC patients with liver metastasis in the real-world. In addition, patients with positive driver genes in our study were excluded to avoid weakening the effectiveness of ICIs.

We further performed a comparison of the response among liver and non-liver lesions from liver metastasis patients who received anti-PD-1/PD-L1 therapy. It was observed that both ORR and DCR in liver lesions were less. The result was in agreement with the conclusions from a recent study, which was done to assess the efficacy among 761 lesions (58 liver metastases) in 214 NSCLC patients treated with PD-1 inhibitor monotherapy and that indicated that liver metastatic lesions showed the worst response [27].

Furthermore, we found that CD8+ T cell infiltration in liver lesions was significantly decreased when compared to non-liver lesions. CD8+ T cells are often considered the important effector cell population for ICI treatment [28,29]. With sufficient cytotoxic lymphocyte infiltration in the tumor microenvironment, blockade of the PD-1/PD-L1 pathway by ICIs can activate effective anti-tumor immunity [28,30]. Therefore, the negative efficacy of PD-1/PD-L1 inhibitors on liver lesions may be due to the lack of CD8+ T cells. As previously mentioned, a minimal proportion of PD-L1+CD8+ ICs in liver metastasis–NSCLC tumors was recorded [13]. Populations who had the limited efficiency from PD-1/PD-L1 inhibitors may owe this to the microenvironment of these populations, and be immunoinflammatory deficient [13,30]. In our data, patients in the PD-L1 and CD8 double-positive expression group had notably longer PFS than those in the single-positive or double-negative expression group. The results suggested that patients whose immune microenvironment exhibits an immunoinflammatory phenotype have better outcomes.

PD-L1 expression is related to an excellent response to anti-PD-1 therapy in NSCLC patients [29,31]. For the contained liver metastasis patients, these who positively expressed PD-L1 had obviously longer PFS. Furthermore, univariate and multifactorial analyses showed that PD-L1 positivity was a significant predictor of PFS. This result indicated that the expression of PD-L1 may play a potentially effective predictor role among liver metastasis-NSCLC patients. This is consistent with the current belief in which PD-L1 could act as a clinically valid prognostic marker for ICIs [32,33,34].

It is supported by previous studies in which it was found that the liver is rich in blood vessels and overexpressed vascular endothelial growth factor (VEGF), which has an immunomodulatory effect [35,36,37,38]. In liver metastasis, anti-angiogenesis therapy such as bevacizumab can block the effect of VEGFR [35,36,37,39]. Thus, anti-angiogenesis therapy may have a positive effect on liver metastasis. A subgroup analysis of patients with liver metastases in Impower150, which is a phase III clinical study, found that the patients receiving atezolizumab combined with chemotherapy plus bevacizumab achieved promising efficacy [5]. Another study about the bevacizumab in NSCLC patients found that bevacizumab significantly improved PFS and OS, and that bevacizumab was more effective in patients with liver metastases than in those with non-liver metastases [40]. However, due to the limited sample among patients with liver metastases in our study, anti-angiogenesis plus PD-1/PD-L1 inhibitors did not display a beneficial advantage (Table A1 and Table A2). Two liver metastasis patients who received PD-1/PD-L1 inhibitors in combination with platinum-containing chemotherapy developed PD. Notably, after received second-line anti-angiogenesis therapy plus chemotherapy, they both had PR (Figure A3). One patient was alive until the follow-up date. The above performance implies that for NSCLC patients with liver metastases, anti-angiogenesis treatment may serve as a promising scheme when they have failed with PD-1/PD-L1 inhibitors; however, further confirmation of this finding in larger scale studies is required.

Data from previous studies have confirmed the poor prognosis of NSCLC patients with liver metastasis [3,4,5]; the potential interventions to improve clinical outcomes are follows. The combination of local treatments such as radiofrequency ablation can be used in patients with liver oligometrics (either a single lesion smaller than 5 cm or as many as three lesions smaller than 3 cm each) [41,42]. PD-1/PD-L1 inhibitors have revealed promising activity in NSCLC patients with liver metastases. The results of the subgroup analysis of IMpower150 showed that the addition of bevacizumab was effective in NSCLC patients with liver metastases, which may be a promising treatment strategy in the future [5,43]. However, in our study, due to the small sample size of liver metastasis patients, expanding the sample size is necessary. Besides, because of the special immune microenvironment of the liver [20], it is essential to seek new immune microenvironment-related therapeutic targets to develop novel therapeutic drugs.

Several limitations in our study. First of all, the sample size of liver metastasis patients was small. Secondly, our study was single-centered and retrospective; besides, we performed PD-L1 testing in NSCLC patients with liver metastases for whom pathological tissue was available, which may account for the difference in PD-L1 expression between patients with and without liver metastases. Therefore, our findings have to be verified in future prospective studies. Finally, adverse events data were not reported because of lack of complete records.

## 5. Conclusions

Our findings imply that PD-1/PD-L1 inhibitors enhances effectiveness in NSCLC patients who have liver metastases, but the benefit is inferior to those in patients without liver metastasis. Furthermore, PD-L1 expression and CD8+ T cell infiltration might act as biomarkers to predict the efficacy of PD-1/PD-L1 inhibitors in patients with NSCLC and liver metastasis.

## Figures and Tables

**Figure 1 cancers-14-04333-f001:**
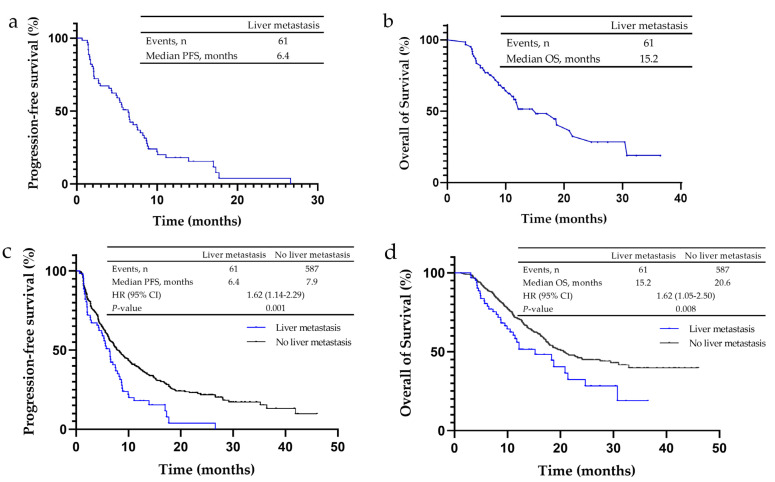
Kaplan–Meier analysis for progression-free survival (PFS) and overall survival (OS) in non-small cell lung cancer (NSCLC) patients treated with PD-1/PD-L1 inhibitors. (**a**) The median PFS of NSCLC patients with liver metastases was 6.4 months. (**b**) The median OS of NSCLC patients with liver metastases was 15.2 months. (**c**) The median PFS of NSCLC patients without liver metastases was 7.9 months, and that of patients with liver metastases was 6.4 months. (**d**) The median OS of NSCLC patients without liver metastases was 20.6 months, and that of patients with liver metastases was 15.2 months.

**Figure 2 cancers-14-04333-f002:**
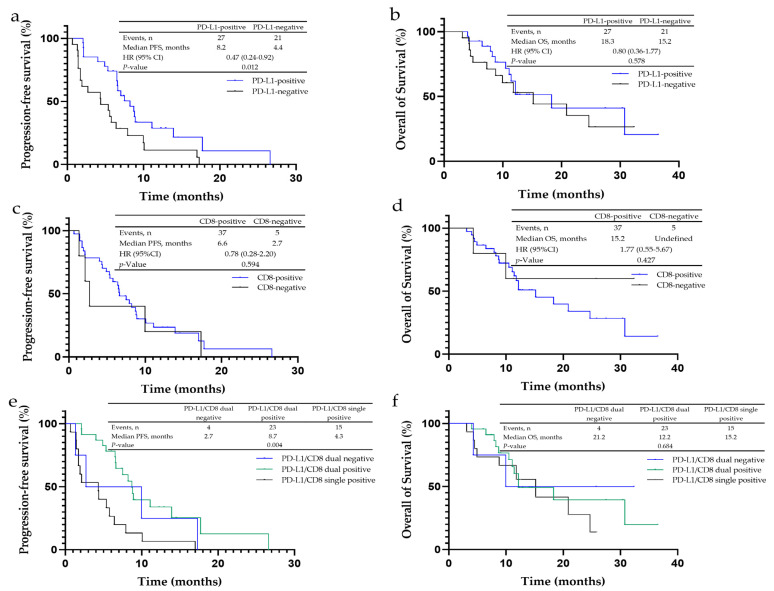
Kaplan–Meier analysis for progression-free survival (PFS) and overall survival (OS) in non-small cell lung cancer (NSCLC) patients with liver metastases to PD-L1, CD4+, CD8+ and CD68+ expression. (**a**) The median PFS of PD-L1-negative cancer was 4.4 months, and that of PD-L1-positive cancer was 8.2 months. (**b**) The median OS of PD-L1-negative cancer was 15.2 months, and that of PD-L1-positive cancer was 18.3 months. (**c**) The median PFS of CD8-negative cancer was 2.7 months, and that of CD8-positive cancer was 6.6 months. (**d**) The median OS of CD8-negative cancer was undefined, and that of CD8-positive cancer was 15.2 months. (**e**) The median PFS of PD-L1/CD8-dual-negative cancer was 2.7 months, that of PD-L1/CD8-dual-positive cancer was 8.7 months, and that of PD-L1/CD8-single-positive cancer was 4.3 months. (**f**) The median OS of PD-L1/CD8-dual-negative cancer was 21.2 months, that of PD-L1/CD8-dual-positive cancer was 12.2 months, and that of PD-L1/CD8-single-positive cancer was 15.2 months.

**Table 1 cancers-14-04333-t001:** Baseline characteristics of included NSCLC patients according to the liver metastasis status.

	Total (*n* = 648)	Liver Metastasis (*n* = 61)	No Live Metastasis (*n* = 587)	*p*-Value
Median age (range), years	63 (32–81)	63 (34–76)	63 (32–81)	
<65	378 (58.3)	36 (59.0)	342 (58.3)	0.909
≥65	270 (41.6)	25 (41.0)	245 (41.7)	
Gender, *n* (%)				
Male	538 (83.0)	55 (90.2)	483 (82.3)	0.119
Female	110 (17.0)	6 (9.8)	104 (17.7)	
ECOG PS, *n* (%)				
0–1	599 (92.4)	53 (86.9)	546 (93.0)	0.085
2	49 (7.6)	8 (13.1)	41 (7.0)	
Smoking status, *n* (%)				
Never	179 (27.6)	11 (18.0)	168 (28.6)	0.078
Current or former	469 (72.4)	50 (82.0)	419 (71.4)	
Histology, *n* (%)				
Adenocarcinoma	329 (50.8)	27 (44.3)	300 (51.1)	0.868
Squamous	270 (44.7)	29 (47.5)	243 (41.4)	
Others	49 (7.6)	5 (8.2)	44 (7.5)	
PD-L1 TPS, *n* (%)				
Unknow	458 (70.7)	13 (21.3)	445 (75.8)	
Know	190 (29.3)	48 (78.7)	142 (24.2)	
Negative	56 (29.5)	21 (43.8)	35 (24.6)	0.012
Positive	134 (70.5)	27 (56.3)	107 (75.4)	
Brain metastases, *n* (%)				
Yes	91 (14.0)	7 (11.5)	84 (14.3)	0.544
No	557 (86.0)	54 (88.5)	503 (85.7)	
Lines of ICI therapy, *n* (%)				
1	316 (48.8)	34 (55.7)	282 (48.0)	0.252
≥2	332 (51.2)	27 (44.3)	305 (52.0)	
ICI treatment regimen, *n* (%)				
ICI monotherapy	250 (38.6)	21 (34.4)	229 (39.0)	0.484
ICI combined therapy	398 (61.4)	40 (65.6)	358 (61.0)	

Abbreviations: ECOG PS, Eastern Corporation Oncology Group performance status; PD-L1, programmed cell death ligand-1; TPS, tumor proportion score; ICI, Immune checkpoint inhibitor.

**Table 2 cancers-14-04333-t002:** Immunotherapy efficacy of NSCLC patients. (**a**) Liver metastases group and non-liver metastasis group. (**b**) Liver and non-liver lesions in NSCLC patients with liver metastasis.

(**a**)		
	**Liver Metastasis** **(*n* = 61)**	**Non-Liver Metastasis** **(*n* = 587)**
CR, *n* (%)	0 (0)	2 (0.3)
PR, *n* (%)	18 (29.5)	208 (35.4)
SD, *n* (%)	26 (42.6)	270 (46.0)
PD, *n* (%)	17 (27.9)	107 (18.2)
ORR, *n* (%)	18 (29.5)	210 (35.8)
DCR, *n* (%)	44 (72.1)	480 (81.8)
(**b**)		
	**Liver lesions**	**Non** **-** **Liver Lesions**
CR, *n* (%)	0 (0)	0 (0)
PR, *n* (%)	16 (26.2)	24 (39.3)
SD, *n* (%)	30 (49.2)	30 (49.2)
PD, *n* (%)	15 (24.6)	7 (11.5)
ORR, *n* (%)	16 (26.2)	24 (39.3)
DCR, *n* (%)	46 (75.4)	54 (88.5)

Abbreviations: CR, complete response; PR, partial response; SD, stable disease; PD, progressive disease; ORR, objective response rate; DCR, disease control rate.

**Table 3 cancers-14-04333-t003:** PD-L1, CD4, CD8, CD68 expression of liver and non-liver lesions in NSCLC patients with liver metastasis.

	Liver Lesions	Non-Liver Lesions	*p*-Value
PD-L1			
Negative	4 (40.0)	17 (44.7)	0.788
Positive	6 (60.0)	21 (55.3)	
CD4			
Negative	3 (30.0)	5 (15.6)	0.312
Positive	7 (70.0)	27 (84.4)	
CD8			
Negative	3 (30.0)	2 (6.3)	0.043
Positive	7 (70.0)	30 (93.8)	
CD68			
Negative	1 (10.0)	7 (21.9)	0.404
Positive	9 (90.0)	25 (78.1)	

Abbreviations: PD-L1, programmed cell death ligand-1; TPS, tumor proportion score.

**Table 4 cancers-14-04333-t004:** Univariate and multivariate analysis of progression-free survival and overall survival in relation to the baseline characteristics of included NSCLC patients with liver metastases. (**a**) Progression-Free Survival. (**b**) Overall Survival.

(**a**)						
**Factor**	**Univariate Analysis**			**Multivariate Analysis**		
	**HR**	**95% CI**	** *p* ** **-Value**	**HR**	**95% CI**	** *p* ** **-Value**
Age (≥65; <65)	1.24	0.70–2.19	0.455			
Gender (female; male)	1.04	0.37–2.91	0.935			
ECOG PS (2; 0–1)	0.63	0.26–1.49	0.289			
Smoking status (current or former; never)	1.17	0.54–2.50	0.695			
Histology						
Adenocarcinoma						
Squamous	0.67	0.37–1.20	0.175			
Others	1.17	0.47–2.91	0.742			
PD-L1 TPS (positive; negative)	0.45	0.24–0.86	0.015	0.39	0.20–0.76	0.006
CD8 (positive; negative)	0.77	0.29–2.01	0.595			
Brain metastases (Yes; No)	2.18	0.97–4.93	0.061	2.86	1.04–7.84	0.041
Treatment line (≥ 2; 1)	1.10	0.64–1.91	0.731			
Treatment regimen (ICI combined therapy; ICIs monotherapy)	0.78	0.44–1.37	0.384			
(**b**)						
**Factor**	**Univariate Analysis**					
	**HR**		**95% CI**		** *p* ** **-Value**	
Age (≥65; <65)	1.38		0.69–2.74		0.359	
Gender (female; male)	0.75		0.23–2.50		0.643	
ECOG PS (2; 0–1)	1.16		0.44–3.01		0.768	
Smoking status (current or former; never)	1.08		0.42–2.82		0.871	
Histology						
Adenocarcinoma						
Squamous	0.56		0.28–1.15		0.113	
Others	0.54		0.12–2.39		0.416	
PD-L1 TPS (positive; negative)	0.80		0.36–1.76		0.579	
CD8 (positive; negative)	1.79		0.42–7.75		0.434	
Brain metastases (Yes; No)	1.81		0.70–4.71		0.223	
Treatment line (≥2; 1)	0.84		0.42–1.67		0.614	
Treatment regimen (ICI combined therapy; ICIs monotherapy)	0.69		0.34–1.39		0.295	

Abbreviations: ECOG PS, Eastern Cooperative Oncology Group performance status; PD-L1, programmed cell death ligand-1; TPS, tumor proportion score; ICI, Immune checkpoint inhibitor.

## Data Availability

The data can be shared up on request.

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
