# Peer review of "The Efficacy of PD-1/PD-L1 Inhibitors in Patients with Liver Metastasis of Non-Small Cell Lung Cancer: A Real-World Study"

_cancers, 2022, doi:10.3390/cancers14174333_

Round 1

Reviewer 1 Report

The manuscript submitted by Xie et al. entitled "The Efficacy of PD-1/PD-L1 Inhibitors in Patients with Liver Metastasis of Non-small Cell Lung Cancer: A Real-world Study" aims to study the advantages of using PD-1/PD-L1 inhibitors in humans with liver metastasis of NSCLC, regarding their PD-L1 status in liver metastasis. Unfortunately, the article and the experimental design have serious flaws that, in my opinion, are very limitative of publishing this work. Indded, the authors don't discriminate which  PD-1/PD-L1 inhibitors used and the therapeutical schemes, making impossible to accept the stratification of the patients. It's critical to stratify the patients not only by the metastatic disease and PD-L1 status but also by each PD-1/PD-L1 inhibitors received. Thus, this article is not acceptable to be published in a journal like Cancers. 

Reviewer 2 Report

The present article entitled, " The efficacy of PD-1/PD-L1 inhibitors in patients with liver metastasis of non-small cell lung cancer: A real world study" by Fan et al., exhibits the relevance and efficiency of ICIs (PD-1/PD-L1) in case study of NSCLC patients.

The sample size, controls, methods and overall information is descriptive and presented in a satisfactory manner. However, minor grammatic changes are required. 

1.  The synonyms for the word efficacy could be used. (the repetitive use of same word in same paragraph could be avoided)

2. If Abbreviations are explained in abstract or anywhere in the article, there is no need to explain/ write the entire term in latter part of the article.

3. Appendix A- Figure A1- Kindly improve the pixel quality of figure. Although overall presentation of data is beautiful in this article, it would be more interesting if this figure uses some creativity as well.

4. Overall the abstract and discussion part is precise in information. However, addition of few more references will be helpful in cross examining and understanding the details presented in this study.

5. The future perspective of this study could use some more information.

6. The work looks promising!

All the best!

Author Response

Response to Reviewer 2 Comments

Point 1: The synonyms for the word efficacy could be used. (the repetitive use of same word in same paragraph could be avoided)

Response: We appreciate this suggestion, and have enriched the vocabulary and checked the grammar of this manuscript in the revised version. This article has been revised by a native English speaker.

Point 2: If Abbreviations are explained in abstract or anywhere in the article, there is no need to explain/ write the entire term in latter part of the article.

Response: We appreciate it very much for this good suggestion. According to your suggestion, we have checked the abbreviations in the abstract and main text in the revised version. According to the style guide of Cancers, “Note that the abstract, main text, and figure/table/scheme captions are treated separately for abbreviations. This means that you need to define the abbreviation the first time you use it in each part—you may have to define the same abbreviation three separate times”, we defined the abbreviations that appeared for the first time in the abstract and the main text, respectively.

Point 3: Appendix A- Figure A1- Kindly improve the pixel quality of figure. Although overall presentation of data is beautiful in this article, it would be more interesting if this figure uses some creativity as well.

Response: Thank you very much for your constructive suggestions. In the revised version, we have modified Appendix A - figure A1, to improve the pixel quality. We have also uploaded the high-definition version of the figure in the separate attachment.

Point 4: Overall the abstract and discussion part is precise in information. However, addition of few more references will be helpful in cross examining and understanding the details presented in this study.

Response: Thank you for the suggestion. We added reference 19 in the revised version to indicate the reasons for liver immune tolerance, reference 30 to indicate the phenomenon through which that ICIs can activate effective anti-tumor immunity by blocking the PD-1/PD-L1 pathway in the tumor microenvironment with sufficient cytotoxic lymphocyte infiltration, reference 34 to indicate that PD-L1 could act as a clinically valid prognostic marker for ICIs, and reference 39 to indicate that anti-angiogenesis therapy can block the effect of VEGFR.

According to the style guide of Cancers that “Abstracts must be self-contained: they are often displayed and read independently of the rest of the paper. This means that any abbreviations used must be defined in the abstract, and no reference can be made to the bibliography or any figures. Citations to previously published papers are not required in abstracts”, we did not cite references in the abstract.

Point 5: The future perspective of this study could use some more information.

Response: Thank you for your suggestion. We have updated the manuscript to include a more comprehensive description of future perspectives (Lines 331-342).

Revision: [Lines 331-342] Data from previous studies have confirmed the poor prognosis of NSCLC patients with liver metastasis [1-3], the potential interventions to improve clinical outcomes are follows. The combination of local treatments such as radiofrequency ablation can be used in patients with liver oligometrics (either a single lesion smaller than 5 cm or as many as three lesions smaller than 3 cm each) [4, 5]. PD-1/PD-L1 inhibitors revealed promising activity in NSCLC patients with liver metastasis. The results of the subgroup analysis of IMpower150 showed that the addition of bevacizumab was effective in NSCLC patients with liver metastasis, which may be a promising treatment strategy in the future [3, 6]. However, in our study, due to the small sample size of liver metastasis patients, further to expand the sample size is necessary. Besides, because of the special immune microenvironment of liver [7], it is essential to seek new immune microenvironment-related therapeutic targets to develop novel therapeutic drugs.

Point 6: The work looks promising!

Response: Thank you for your kind words regarding our manuscript.

Point 7: Line #30- Why and how? Kindly cite the references.

Line # 53, 59 and 60 - repetitive explanation of abbreviations is not necessary.

Line # 357: Poor quality Figure.

Line # 383: Kindly cite the references properly (Abstract and Introduction sections)

Response: Thank you very much for your constructive suggestions. We modified Figure 1, Figure 2 and Appendix A - figure A1 for improve quality in the revised manuscript. We have also uploaded the high-definition version of these figures in as the separate attachment.

According to the style guide of Cancers that “Note that the abstract, main text, and figure/table/scheme captions are treated separately for abbreviations. This means that you need to define the abbreviation the first time you use it in each part—you may have to define the same abbreviation three separate times”, we have revised the manuscript to only define abbreviations at their first appearance in the abstract and main text.

Line #30 is part of the abstracts, according to “Abstracts must be self-contained: they are often displayed and read independently of the rest of the paper. This means that any abbreviations used must be defined in the abstract, and no reference can be made to the bibliography or any figures. Citations to previously published papers are not required in abstracts”, we did not cite references in the abstract.

References

[1] S. Gadgeel, D. Rodriguez-Abreu, G. Speranza, E. Esteban, E. Felip, M. Domine, R. Hui, M.J. Hochmair, P. Clingan, S.F. Powell, S.Y. Cheng, H.G. Bischoff, N. Peled, F. Grossi, R.R. Jennens, M. Reck, E.B. Garon, S. Novello, B. Rubio-Viqueira, M. Boyer, T. Kurata, J.E. Gray, J. Yang, T. Bas, M.C. Pietanza, M.C. Garassino, Updated Analysis From KEYNOTE-189: Pembrolizumab or Placebo Plus Pemetrexed and Platinum for Previously Untreated Metastatic Nonsquamous Non-Small-Cell Lung Cancer, J. Clin. Oncol. 38(14) (2020) 1505-1517.

[2] E.E. Vokes, N. Ready, E. Felip, L. Horn, M.A. Burgio, S.J. Antonia, O. Aren Frontera, S. Gettinger, E. Holgado, D. Spigel, D. Waterhouse, M. Domine, M. Garassino, L.Q.M. Chow, G. Blumenschein, Jr., F. Barlesi, B. Coudert, J. Gainor, O. Arrieta, J. Brahmer, C. Butts, M. Steins, W.J. Geese, A. Li, D. Healey, L. Crino, Nivolumab versus docetaxel in previously treated advanced non-small-cell lung cancer (CheckMate 017 and CheckMate 057): 3-year update and outcomes in patients with liver metastases, Ann. Oncol. 29(4) (2018) 959-965.

[3] N. Nogami, F. Barlesi, M.A. Socinski, M. Reck, C.A. Thomas, F. Cappuzzo, T.S.K. Mok, G. Finley, J.G. Aerts, F. Orlandi, D. Moro-Sibilot, R.M. Jotte, D. Stroyakovskiy, L.C. Villaruz, D. Rodriguez-Abreu, D. Wan-Teck Lim, D. Merritt, S. Coleman, A. Lee, G. Shankar, W. Yu, I. Bara, M. Nishio, IMpower150 Final Exploratory Analyses for Atezolizumab Plus Bevacizumab and Chemotherapy in Key NSCLC Patient Subgroups With EGFR Mutations or Metastases in the Liver or Brain, J. Thorac. Oncol. 17(2) (2022) 309-323.

[4] F. Izzo, V. Granata, R. Grassi, R. Fusco, R. Palaia, P. Delrio, G. Carrafiello, D. Azoulay, A. Petrillo, S.A. Curley, Radiofrequency Ablation and Microwave Ablation in Liver Tumors: An Update, Oncologist 24(10) (2019) e990-e1005.

[5] Y. Minami, N. Nishida, M. Kudo, Radiofrequency ablation of liver metastasis: potential impact on immune checkpoint inhibitor therapy, Eur. Radiol. 29(9) (2019) 5045-5051.

[6] M. Reck, T.S.K. Mok, M. Nishio, R.M. Jotte, F. Cappuzzo, F. Orlandi, D. Stroyakovskiy, N. Nogami, D. Rodríguez-Abreu, D. Moro-Sibilot, C.A. Thomas, F. Barlesi, G. Finley, A. Lee, S. Coleman, Y. Deng, M. Kowanetz, G. Shankar, W. Lin, M.A. Socinski, M. Reck, T.S.K. Mok, M. Nishio, R.M. Jotte, F. Cappuzzo, F. Orlandi, D. Stroyakovskiy, N. Nogami, D. Rodríguez-Abreu, D. Moro-Sibilot, C.A. Thomas, F. Barlesi, G. Finley, A. Lee, S. Coleman, Y. Deng, M. Kowanetz, G. Shankar, W. Lin, M.A. Socinski, Atezolizumab plus bevacizumab and chemotherapy in non-small-cell lung cancer (IMpower150): key subgroup analyses of patients with EGFR mutations or baseline liver metastases in a randomised, open-label phase 3 trial, The Lancet Respiratory Medicine 7(5) (2019) 387-401.

[7] A. Limmer, J. Ohl, C. Kurts, H.G. Ljunggren, Y. Reiss, M. Groettrup, F. Momburg, B. Arnold, P.A. Knolle, Efficient presentation of exogenous antigen by liver endothelial cells to CD8+ T cells results in antigen-specific T-cell tolerance, Nat Med 6(12) (2000) 1348-54.

Reviewer 3 Report

This current work by Xie et al., validates the efficacy of  PD-L1 inhibitors in NSCLC patients with and without liver metastasis. Moreover, they found PD-L1 inhibitors are more effective in  NSCLC patients with liver metastasis. In addition to that they implicated CD8+ T cell could serve as potential biomarkers for PD-L1 inhibitors therapy  in NSCLC patients with liver metastasis. 

 I am in principle supportive of accepting this work for publication. However, I have few suggestions to improve the manuscript for publication. 

Major 

  1. I strongly encouraged authors to quantitate the expression of PDL-1 in the tissue section in Figure A2. 

Minor 

  1. Table in Figure 1 and Figure 2 should be legible

Author Response

Response to Reviewer 3 Comments

Point 1: I strongly encouraged authors to quantitate the expression of PD-L1 in the tissue section in Figure A2. 

Response: According to your suggestion, we have quantified the expression of PD-L1 in the tissue section in Figure A2 in the revised version (Lines 380-381). Revision: [Lines 380-381] (a) Positive expression of PD-L1 (PD-L1 TPS=40%); (b) negative expression of PD-L1 (PD-L1 TPS < 1%).

Point 2: Table in Figure 1 and Figure 2 should be legible

Response: Thank you very much for your suggestions. We have improved the clarity of the table in Figure 1 and Figure 2 in the revised version. We have also uploaded the high-definition version of these figures in the separate attachment.

Round 2

Reviewer 1 Report

This study has several limitions as the very low sample number, patients were not not stratified by molecular tumor subtype and many therapetic protocols were used.